# Assessing Face Validity of the HexCom Model for Capturing Complexity in Clinical Practice: A Delphi Study

**DOI:** 10.3390/healthcare9020165

**Published:** 2021-02-04

**Authors:** Xavier Busquet-Duran, Eva Maria Jiménez-Zafra, Magda Tura-Poma, Olga Bosch-de la Rosa, Anna Moragas-Roca, Susana Martin-Moreno, Emilio Martínez-Losada, Silvia Crespo-Ramírez, Lola Lestón-Lado, Núria Salamero-Tura, Joana Llobera-Estrany, Núria Oriol-Peregrina, Eduard Moreno-Gabriel, Josep Maria Manresa-Domínguez, Pere Torán-Monserrat

**Affiliations:** 1Home Care Program, Granollers Support Team (PADES), Vallès Oriental Primary Care Service, Catalan Health Institute, 08520 Granollers, Spain; evajiza@gmail.com (E.M.J.-Z.); turapoma@hotmail.com (M.T.-P.); amoragas_roca@hotmail.com (A.M.-R.); susmartin0@hotmail.com (S.M.-M.); emlosada1974@gmail.com (E.M.-L.); lestonlola@gmail.com (L.L.-L.); nsalamero.mn.ics@gencat.cat (N.S.-T.); jolloes@hotmail.com (J.L.-E.); 2Multidisciplinary Research Group on Health and Society (GREMSAS), (2017 SGR 917), 08007 Barcelona, Spain; emoreno@idiapjgol.info (E.M.-G.); jmanresa@idiapjgol.info (J.M.M.-D.); ptoran.bnm.ics@gencat.cat (P.T.-M.); 3Nursing Department, Fundació Universitària Bages (FUB), University of Vic, 08500 Vic, Spain; 4Red Cross Psychosocial Care Team (EAPS), 08402 Granollers, Spain; olga.bosch@creuroja.org (O.B.-d.l.R.); silvia.crespo@creuroja.org (S.C.-R.); 5Degree in Speech and Language Therapy, University of Vic-Central University of Catalonia/UOC, 08242 Manresa, Spain; noriol@umanresa.cat; 6Sociosanitari Vallparadís, 08221 Terrassa, Spain; 7Research Support Unit Metropolitana Nord, Primary Care Research Institut Jordi Gol (IDIAPJGol), 08303 Barcelona, Spain; 8Nursing Department, Faculty of Medicine, Universitat Autònoma de Barcelona, 08193 Barcelona, Spain

**Keywords:** palliative care, Delphi technique, needs assessment, home care services, coordinated care, complexity, methodological study

## Abstract

Capturing complexity is both a conceptual and a practical challenge in palliative care. The HexCom model has proved to be an instrument with strong reliability and to be valid for describing the needs and strengths of patients in home care. In order to explore whether it is also perceived to be helpful in enhancing coordinated and patient-centred care at a practical level, a methodological study was carried out to assess the face validity of the model. In particular, a Delphi method involving a group of 14 experts representing the full spectrum of healthcare professionals involved in palliative care was carried out. The results show that there is a high level of agreement, with a content validity index-item greater than 0.92 both with regard to the complexity model and the HexCom-Red, HexCom-Basic, and the HexCom-Clin instruments, and higher than 0.85 regarding the HexCom-Figure and the HexCom-Patient instruments. This consensus confirms that the HexCom model and the different instruments that are derived from it are valued as useful tools for a broad range of healthcare professional in coordinately capturing complexity in healthcare practice.

## 1. Introduction

Due to the inclusion of all groups of pathologies within the scope of palliative care [1,2] and the incessant aging of the population [3], substantial growth is expected in the demand for these healthcare services in the upcoming years [4]. In the Spanish context, it is considered that 1.5% of the population and 11% of people over 65 need palliative care [5,6,7]. This progressive and quantitative increase brings along a sophistication of this demand. Accordingly, experts agree that the backbone of palliative care must be the notion of complexity [8,9,10,11]. However, capturing complexity is both a conceptual and a practical challenge in palliative care [11].

The very nature of “complexity” is controversial [12]. This is partly due to the fact that it emerges from the concurrence of multiple interrelated factors, whether they are clinical, contextual, or related with the health system [13]. In an attempt to enhance the concept’s applicability and endorse a patient-centred clinical practice, a number of systemic theoretical frameworks have been developed and adopted over the last few years in the field of palliative care. The cumulative model puts the focus on the imbalance between the burdens and responsibilities of the patient on the one hand and their capacities on the other [14]. The vector model focuses on contextual factors, namely the interrelation and balance between socioeconomic, behavioural, genetic, environmental, cultural, and sociopolitical factors [5,15,16]. Finally, drawing on Bronfenbrenner’s ecological systems theory, Pask et al. [11] offer an innovative framework for understanding complexity by emphasizing the natural ecological environment of the individual, which is formed by the set of relationship structures that accommodate him/her.

This systemic approach is particularly relevant for the practice of home care in Spain. In contrast with how palliative care is delivered at other, more institutionalised levels of care, foregrounding the patient’s specific needs and seeing them in relation to a broader and specific context should facilitate providing timely, adequate, individualised, and coordinated care [12,17]. Likewise, managing patients’ profiles drawing on their specific needs should assist in identifying those in need of specialised palliative care [11,18]. Additionally, this holistic perspective enhances home care teams in their responsibility to act as coordinators of different resources and levels of care involved in palliative care in Spain [19].

To the best of our knowledge, there are not available validated instruments based on this systemic approach, neither at a national level [20,21,22,23,24] nor at the international level [25,26,27,28,29,30,31,32,33,34,35]. To overcome this deficit, between 2006 and 2017 a model (HexCom) was designed to ultimately aid healthcare professionals caring for people with advanced illness and/or at the end of life.

Originally, this model drew on the complexity levels and the inclusion and referral criteria for palliative care patients defined by the Catalan Government drawing on the conclusions of an interdisciplinary and multilevel work group endorsed by the Catalan Scientific Societies for Palliative Care and Family Medicine and Community [36]. Based on Ferris [37], the model proposes six areas of needs: clinical, psychoemotional, social/family, spiritual, ethical, and death-related. For each area, the level of complexity may be categorised as low (L), medium (M), or high (H). 

Recently, based on Pask et al. [11], the model has been complemented adding five areas of systemic strength: microsystem, mesosystem, chronosystem, exosystem and macrosystem. For each system, the strength level can also be qualified as low, medium, or high. As a result of contrasting identified needs and strengths, an intuitive balance of complexity is proposed, emerging from considering whether a large amount of systemic strength may or may not reduce the resulting degree of complexity. The model has been published [38] and its content partially validated [39], with an interobserver Kappa of 0.92. The first results of its application in clinical practice have also been published recently [40].

Even though referrer professionals in hospitals and primary care are ultimately in charge of detecting cases with complex needs, palliative care is a basic responsibility of all healthcare professionals [12]. Accordingly, HexCom is deployed in five instrumental versions, which are complementary, suited to the degree of specialization of the service using it and the objective pursued (see Table 1 and Appendix A).

In the Spanish context, specialist palliative care teams generally comprise physicians, nurses, social workers, and psychologists. However, in some areas, these can occasionally include professionals from other backgrounds, such as physiotherapy or even pain and pastoral care [19]. Given this diversity of tools and professionals involved in a complexity assessment, besides the need to achieve coordinated and patient-centred care in palliative care, it is critical to evaluate whether these instruments are consensually perceived, by a broad range of healthcare professionals, as valuable resources for their job of capturing the degree of complexity of all patients’ needs [41]. Hence, this study aims at assessing the face validity of both the Catalan and Spanish versions of the model and its derived instruments.

## 2. Materials and Methods

This study has been conducted and reported following the recommendations for the conducting and reporting of Delphi studies (CREDES) [42].

### 2.1. Design of the Study

The present is a methodological study using a Delphi process to systematically evaluate the face validity of HexCom model and its five instruments. 

Delphi is a technique for determining the level of formal consensus and is used to obtain and evaluate the opinions of a group of experts with knowledge and experience in a specialised field [43,44,45]. In this case, the participants are considered experts in so far as they have knowledge and experience in delivering palliative care, therefore being capable of assessing whether the HexCom model is appropriate to the targeted construct (i.e., complexity) and assessment objectives (i.e., to identify complex patients/situations and allocate resources accordingly) [46]. The method structures a group communication process that is effective in allowing a group of individuals, as a whole, to address a complex issue [47]. The qualitative and quantitative approaches are combined into a single procedure, mitigating their respective limitations [48]. It entails the administration of questionnaires in different rounds until consensus is reached [43]. This method guarantees anonymity, controlled feedback, and a systematic assessment of face validity by way of an iterative process and the use of statistical techniques [49,50].

The procedure, including the previous stages of the HexCom model development, is shown in Figure 1. The two stages belonging to the Delphi study reported in this article are detailed below: (1) the preparation of the questionnaire by a coordinator-administrator group, and (2) the consensus methodology with a panel of experts.

### 2.2. Preparation of the Questionnaire

The coordinator–administrator group was formed by the core members of PADES in Granollers, a home and palliative care support team whose composition is paradigmatic of this type of team in Catalunya (Spain). Namely, it consists of a majority of female professionals with different backgrounds and a longstanding experience in palliative care and home care (see Table 2 for a detailed description).

Drawing on their expertise and the existing literature [51,52], the coordinator–administrator group defined the panel of experts, designed the assessment questionnaire, and provided feedback to the panel.

Based on Oriol-Peregrina [53], the questionnaire contains a total of 60 statements or questions regarding the form, content, utility, and clarity of the HexCom model and its different versions (see Table A1). More specifically, questions refer to the model (questions 2–15), Red (q. 16–23), Basic (q. 24–32), Clin (q. 33–38), Figure (q. 39–45), Patient (q. 46–59), the clarity of the translation into Spanish, its correspondence with the original Catalan version (60), and one last open question to collect generic qualitative feedback on the model or any of the instruments. Participants were asked to provide their degree of agreement with each statement using a Likert scale including the following grades: 1 = “Strongly disagree”, 2 = “Disagree”, 3 = “Indifferent”, 4 = “Agree”, and 5 = “Strongly agree”. All items included a blank space to allow explanatory comments and suggested modifications (e.g., alternative wording of a given item).

### 2.3. Delphi Process

The Delphi technique has been widely used in palliative care [42]. The first areas where it was used were those of oncology pain [45], and it has been used specifically to assess complexity [54].

#### 2.3.1. Participants and Setting

The number of participating experts recommended in the literature is between 7 and 30 [55,56]. Due to the small number of participants, the method is not intended to produce statistically significant results but to collect the synthesis of thought from a special group [49].

The panel of experts was assembled using a purposive sampling process following the selection criteria of their knowledge and experience in palliative and home care at all levels of care (i.e., primary, secondary, etc.), as well as making sure it included those professions most often involved in end-of-life care. Hence, as in the coordinator–administrator groups and generally in the palliative and home care professions in Spain [40], the resulting panel comprised mostly female professionals and did not include professional profiles which are part of palliative care teams on very rare occasions (e.g., physiotherapists, speech therapists, and occupational therapists). Following the established recommendations [40,41] for this procedures, 15 people were selected, with the additional inclusion criteria of having over 7 years of experience in palliative care. Criteria for exclusion consisted of being currently unemployed, retired, or exempt of actual clinical practice for any given reason and having participated in the development of the HexCom model or any of its instruments.

#### 2.3.2. Data Collection

After the questionnaire was designed between January and March 2019, the experts were reached out to in March 2019 in order to present them with the project and ask for their participation. In case of acceptance, a personalised explanatory letter was sent by email, including a link to access the virtual anonymous platform containing complementary documentation, the different versions of the instrument (see Appendix A), and the questionnaire. 

#### 2.3.3. Ethical Considerations

The participants were informed about the project’s objectives, indicating that their participation was voluntary and that the information was collected in an anonymous database so they could not be identified. The study was approved by the Ethics Committee of the IDIAP Fundació Jordi Gol (Ref. P15/171), the official institutional review board for primary care research in Catalunya.

#### 2.3.4. Data Analysis

To assess the degree of consensus after each round, the scores were analysed using the content validity by item index (CVI-I) and the number of experts that showed some disagreement. CVI-I allows assessing quantitatively the degree of agreement amongst experts on a given item, dividing the number of experts agreeing (scoring 4 or 5) with a given item by the total number of experts responding to that question [57]. Following the American Educational Research Association recommendations for studies with a panel of 6 or more experts, a CVI-I of over 0.78 was interpreted as significant consensus [58,59]. After obtaining and analysing the results of the previous round, those items with a CVI-I under 0.78 and/or where three or more experts expressed disagreement were resubmitted in a second and final round [38,60]. Based on Varela et al. [61], the qualitative data obtained in the first round were used in the second round to modify the item in question (see Table A2).

## 3. Results

### 3.1. Description of the Panel of Experts

A description of the panel of experts whose participation was sought is displayed in Table 3. Fourteen of the 15 invited professionals agreed to take part of the study (93%). The average age of the panel was 52.3 years, with all professions at all levels represented, extensive work experience on average was 24.8 years, including long experience both in palliative care (18.1 years) and home care (13 years), as well as having teaching and research experience in the subject (see Table 3).

### 3.2. Delphi Survey

The results of the first and the second Delphi rounds are shown in Table 4. Even though in 54 of the items (90%) the defined threshold for assuming consensus was achieved in the first round, all of those which received critical commentaries or appropriate suggested modifications (15 items) were revised, modified, and resubmitted for validation in the second round, alongside those six items scoring suboptimal CVI-I (<0.78) in the first round. All items scored above 0.78 CVI-I in the second and final round.

#### 3.2.1. The HexCom Model

As shown in Table 4, all items (2–15 and 60) assessing the model and its perceived usefulness reached the established degree of consensus in the first round (CVI-I > 0.86) which increased in the second round (CVI-I > 0.92) after modifying four items drawing on the commentaries received from the panel. According to the qualitative feedback received, this very high consensus stems mainly from two key elements. On the one hand, the sound, conceptual underpinnings of the model: that complexity must be based on changing needs as the backbone of palliative care and the usefulness of a systemic approach to evaluate both needs and strengths in a given case. On the other hand, though slightly less significant, the model’s perceived transferability to all levels of care and, ultimately, the general public.

#### 3.2.2. HexCom Instruments

After adjusting those formal aspects (Item 18) suggested by the panel between rounds, HexCom-Red exceeded 0.92 in CVI-I for all items in the second round, except in the item (21) relating to the explanatory wording. Nevertheless, agreement upon this item improved from 0.79 in the first round to 0.86 in the second.

Similarly, HexCom-Basic exceeded 0.92 in the CVI-I scores for almost all items in the second round (improving the scores in three of four items modified and resubmitted in the second round). Item 29, which refers to the explanatory wording for each area of systemic strengths is an exception to this trend. This scored a CVI-I of 0.78 in the first round and did not improve in the second round. This was partly due to the fact that three experts responded “indifferent” in the first round but did not provide alternatives to the wording.

All items regarding HexCom-Clin achieved the established degree of consensus, scoring 0.86 or above in the CVI-I of all items considering both round 1 and 2. After registering insufficient degree of consensus in the first round, Item 35 (referring to the instrument’s format) was modified and validated in the second round, where it scored a CVI-I of 0.86. Item 38, asking about non-specialists’ understanding of the instrument was resubmitted with modifications in the second round, where it scored 0.86 in CVI-I, as in the first round.

In the first round, the panel of experts assigned rather low scores to formal aspects (Item 41) and content issues (Item 42 and 44) of the HexCom-Figure. However, after the suggested modifications were introduced between rounds, all items exceeded 0.85 of the CVI-I score.

All questions about the HexCom-Patient received were agreed upon in the first round, scoring CVI-I’s over 0.85. Even though two items were resubmitted in the second round after introducing minor modifications to the instruments, the overall degree of agreement increased slightly (one item reached a CVI-I of 1).

#### 3.2.3. Open Comments

The majority of the responses given to the final open question did not refer to the model or the instruments’ contents but to the difficulties of implementing them in non-specialised environments. These included comments on great care pressure, time constrains (both at primary care and hospital levels), fragmented work dynamics, and, as one expert put it, “the need for awareness, information, and specific training and practice”. In sum, the HexCom model is perceived as a good instrument for research in as much as “the model proposes an approach from ‘the whole’, taking into account all aspects that need to be considered, which facilitates monitoring and decision making”. Some experts pointed out the “perils” of forgetting that, before administering the instruments, it is necessary to create a bond and respond to emerging needs.

## 4. Discussion

Establishing a consensus on the complexity of care in a field of clinical practice such as palliative care is not exempt from difficulties [62]. Aspects such as the elements that make it up, the criteria that define it, the best tools for detection and management or its adaptation to a certain healthcare field are still controversial [39]. Thus, studies using the Delphi method in palliative care [44,63] reach levels of agreement lower than in other fields, where up to 100% consensus is achieved in all items in the second round.

In our study, a group of highly qualified experts in palliative care reached an agreement that exceeds 92% with respect to the face validity of the HexCom model and more than 86% in relation to the utility of the different instruments that result from it.

It can be argued that this high degree of consensus is associated with the systemic approach [11] underlying the HexCom model, which allows the detection of unmet needs and understands complexity not only as a multidimensional construction, but also a multireferential construction, taking into account the training of the diverse healthcare teams involved in complexity assessment [34].

The HexCom model is aligned with the majority of instruments validated both internationally [25,26,27,28,29,32], as well as in the context of Spain [33], yet its high perceived usability may be explained by its capacity to broad the scope of application beyond oncological and hospitalised patients [37,38,39]. However, as the panel noted in their open comments, implementing the model in clinical environments characterised by considerable time constrains, amongst other barriers, is certainly challenging. Exploring efficient ways to integrate the model in these settings, taking into account not only “experts” but also potential users’ perspectives, emerges as a research and policy priority.

Arguably, HexCom is appreciated by a wide range of professionals in so far as it improves the responsiveness of instruments such as the PCSPS (palliative care problem severity score), which only considers four areas and refers to features closer to assessing severity rather than complexity [26]. It can also be argued that HexCom is perceived to present certain advantages with respect to the needs assessment tool (NAT: PD-C) [29,31,32] since the former introduces a specific ethical area. Given that the HexCom model identifies complexity on every case where at least one of its dimensions is defined as highly complex, it can be argued that it is valued as more sensitive than similar tools such as PALCOM [33,34], which identifies a patient’s complexity only as a result of a summation of all of its areas. Similarly, it may be perceived as more patient-oriented than IDC-PAL^®^ in as much as it is based on patients’ needs, rather than a set of risk factors [20,21,22].

### Limitations of the Study

A limitation in the Delphi technique is the bias risk in selecting the group of experts. The homogenization criterion (having expertise in home care) may have produced an approach with little representation, potentially limiting the breadth of experience included in the panel. To overcome this threat, the coordinating group consisted of a multidisciplinary group, with extensive and divergent experience, including psychologists and social workers, who are often not represented in other validation processes. However, the elevated interest in participating in the study (93% of the invitees participated) suggests that the participants are very representative of this professional community [56,64]. Even though heterogeneity between this sample was sought by including the different professions involved in home care, it could have been greater if patients and relatives had been somehow involved in the study [63]. Another potentially problematic issue when carrying out Delphi studies is the way in which consensus is defined and operationalised, since it varies considerably amongst different studies [60], particularly in face-validity studies. To address this limitation, a quantitative criteria and established recommendations were followed (CVI-I and the number of experts in disagreement) combined with qualitative data to understand disagreement and enhance the consistency of the final results [47]. 

## 5. Conclusions

This study stems from the need to establish a shared understanding and widely accepted tools for capturing complexity as a critical step towards enacting coordinated and patient-oriented palliative care [33]. At a conceptual level, the HexCom model fulfils the prerequisites proposed by Pask, outlined drawing on patients’ and professionals’ experiences, in order to capture complexity [11]. These entail addressing it systematically; understanding the person as a whole; addressing it at the patient level (in a personal clinical visit and collecting clinical data); identifying needs and resources (resilience and support networks included); defining levels of complexity; and using sensitive instruments that facilitate concrete interventions. This study complements the model’s robustness by validating HexCom face validity for the broad majority of its targeted end-users, namely healthcare professionals from different backgrounds. These positive results consolidate the broader project of validating this model, which will be followed by assessing the test–retest reliability, validity of criteria, concomitant and predictive reliability, as well as sensitivity and feasibility [65]. Accordingly, HexCom model is expected to be increasingly and widely adopted as a shared and valuable tool for needs assessment in palliative care, posing new challenges both for the model’s implementation and established practices and existing limitations for delivering optimal palliative care.

## Figures and Tables

**Figure 1 healthcare-09-00165-f001:**
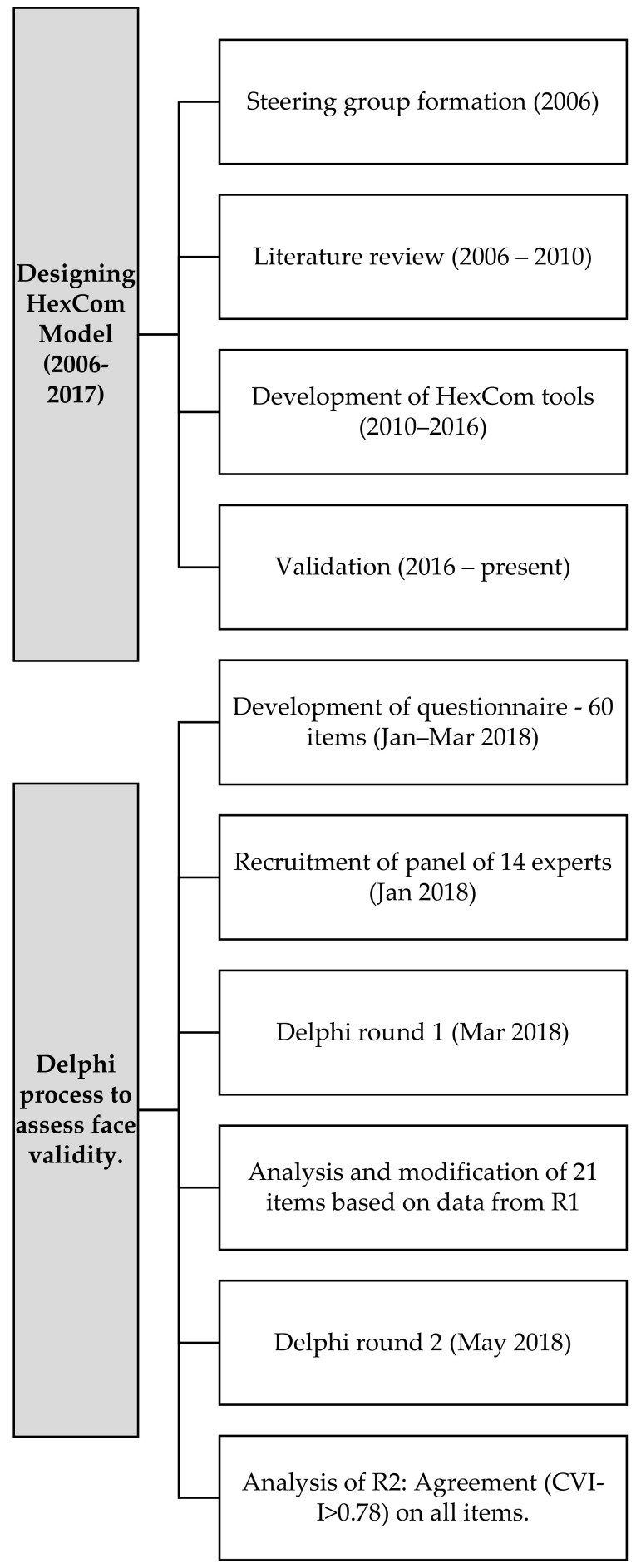
Process of development and validation of HexCom.

**Table 1 healthcare-09-00165-t001:** Instruments derived from the HexCom model.

HexCom	Format	Needs	Strengths	HC Professional End-Users
Red	Table	6 areas	-	Generalists.
Basic	Table	6 areas	5 systems	Specialists with basic equipment.
Clin	Table	6 areas subdivided into 18 subareas	5 systems subdivided into 9 subsystems	Specialists With full set-up.
Figure	Figure	6 areas subdivided into 18 subareas + evaluation at the beginning	5 systems subdivided into 9 subsystems	Specialists with complete resources & training.
Patient	Table	6 areas + 1 open question	5 areas + 2 open questions	Specialists.

**Table 2 healthcare-09-00165-t002:** Characteristics of the coordinating group.

Coordinator Group (*n* = 11)	
**Age**	
From 29 to 45 years	8 (72.7%)
From 45 to 63 years	3 (27.2%)
Average (Standard deviation)	45.8 (8.8)
**Gender**	
Women	9 (81%)
Men	2 (19%)
**Profession**	
Nursing	4 (36.3%)
Medicine	3 (27.2%)
Social work	2 (18.1%)
Psychology	2 (18.1%)
**Years of Professional Experience**	
From 7 to 10	2 (18.1%)
From 11 to 20	4 (36.3%)
From 21 to 39	5 (45.4%)
Average (Standard deviation)	19.3 (8.0)
**Years of Palliative Experience**	
From 7 to 10	2 (18.1%)
From 11 to 20	6 (54.5%)
From 21 to 39	3 (27.2%)
Average (Standard deviation)	17.1 (8.9)
**Years of Home Care Experience**	
From 1 to 10	6 (54.5%)
From 11 to 20	2 (18.1%)
From 21 to 39	3 (27.2%)
Average (Standard deviation)	13.6 (10.8)
**Experience in Teaching and Research**	
Graduate teaching	6 (54.5%)
Postgraduate teaching	7 (63.6%)
Publications	9 (81.8%)

**Table 3 healthcare-09-00165-t003:** Characteristics of the expert panel.

Panel of Experts (*n* = 14)	
**Age**	
From 29 to 45 years	4 (28.6%)
From 45 to 63 years	10 (71.4%)
**Sex**	
Female	10 (71.4%)
Male	4 (28.6%)
**Profession**	
Nursing	3 (21.4%)
Medicine	5 (35.8%)
Social work	3 (21.4%)
Psychology	3 (21.4%)
**Place of Current Work**	
Social healthcare	4 (28.6%)
Primary care	3 (21.4%)
PADES	6 (42.8%)
University	1 (7.1%)
**Years of Professional Experience**	
From 7 to 10	0 (0%)
From 11 to 20	5 (35.7%)
From 21 to 39	9 (64.3%)
**Years of Palliative Experience**	
From 7 to 10	4 (28.6%)
From 11 to 20	3 (21.4%)
From 21 to 39	7 (50%)
**Years of Home Care Experience**	
From 1 to 10	8 (57.1%)
From 11 to 20	1 (6.6%)
From 21 to 39	5 (33.3%)
**Teaching and Research Experience**	
Graduate	8 (57.1%)
Postgraduate	12 (85.7%)
Publications	13 (92.8%)

**Table 4 healthcare-09-00165-t004:** Expert answers rounds 1 and 2, with average, standard deviation (SD), number of experts in disagreement, and the content validity by item index (CVI-I).

Item	Round 1				Round 2			
	*N*	Average (SD)	Experts not in Agreement (*N*)	CVI-I	*N*	Average (SD)	Experts not in Agreement (*N*)	CVI-I
**Model**								
2	14	4.71 (0.47)	0	1	14			
3	14	4.71 (0.47)	0	1	14	4.47 (1.06)	0	1
4	14	4.71 (0.47)	0	1	14			
5	14	4.86 (0.36)	0	1	14			
6	14	4.21 (0.58)	1	0.93	14			
7	14	4.71 (0.47)	0	1	14			
8	14	4.57 (0.51)	0	1	14			
9	14	4.57 (0.51)	0	1	14			
10	14	4.57 (0.51)	0	1	14			
11	14	4.29 (0.61)	1	0.93	14			
12	14	4.36 (0.50)	0	1	14			
13	14	3.86 (0.66)	2	0.86	14	3.80 (0.88)	1	0.93
14	14	4.50 (0.52)	0	1	14	4.13 (0.99)	0	1
15	14	4.14 (0.66)	2	0.86	14	3.87 (0.83)	0	1
**HexCom-Red**								
16	14	4.64 (0.50)	0	1	14			
17	14	4.36 (0.63)	1	0.93	14			
18	14	3.93 (0.73)	4	0.71	14	3.93 (1.11)	1	0.93
19	14	4.14 (0.53)	1	0.93	14			
20	14	4.14 (0.53)	1	0.93	14			
21	14	4.14 (0.77)	3	0.79	14	3.92 (1.06)	2	0.86
22	14	4.36 (0.93)	2	0.86	14	4.40 (1.07)	1	0.93
23	14	4.21 (0.43)	0	1	14			
**HexCom-Basic**								
24	14	4.57 (0.51)	0	1	14			
25	14	4.29 (0.83)	1	0.93	14			
26	14	4.14 (0.86)	2	0.86	14	3.86 (1.07)	1	0.93
27	14	4.21 (0.58)	1	0.93	14			
28	14	3.93 (1.00)	3	0.79	14	3.86 (1.07)	1	0.93
29	14	4.00 (0.68)	3	0.78	14	3.85 (1.04)	3	0.78
30	14	4.14 (0.53)	1	0.93	14			
31	14	4.14 (0.77)	1	0.93	14			
32	14	4.00 (0.96)	4	0.71	14	3.87 (0.83)	0	1
**HexCom-Clin**								
33	14	4.57 (0.51)	0	1	14			
34	14	4.36 (0.84)	1	0.93	14			
35	14	3.57 (1.16)	5	0.64	14	3.92 (1.13)	2	0.86
36	14	4.29 (0.47)	0	1	14			
37	14	4.00 (0.78)	2	0.86	14	3.93 (0.88)	0	1
38	14	4.14 (0.86)	2	0.86	14	3.79 (0.90)	2	0.86
**HexCom-Figure**								
39	14	4.57 (0.51)	0	1	14			
40	14	4.29 (0.47)	0	1	14			
41	14	3.93 (0.92)	4	0.71	14	3.66 (0.85)	2	0.86
42	14	3.79 (0.89)	5	0.64	14	3.87 (0.83)	0	1
43	14	4.07 (0.27)	0	1	14	3.87 (0.83)	0	1
44	14	3.64 (0.84)	4	0.71	14	3.72 (0.83)	2	0.86
45	14	4.36 (0.63)	1	0.93	14			
**HexCom-Patient**								
46	14	4.50 (0.52)	0	1	14			
47	14	4.50 (0.52)	0	1	14			
48	14	4.29 (0.61)	1	0.93	14			
49	14	4.36 (0.63)	1	0.93	14			
50	14	4.21 (0.80)	1	0.93	14	3.93 (0.88)	0	1
51	14	4.29 (0.47)	0	1	14			
52	14	4.29 (0.47)	0	1	14			
53	14	4.29 (0.61)	1	0.93	14			
54	14	4.15 (0.80)	1	0.92	14			
55	14	4.21 (0.70)	2	0.86	14	4.12 (1.15)	2	0.86
56	14	4.71 (0.47)	0	1	14			
57	14	4.86 (0.36)	0	1	14			
58	14	4.14 (0.53)	1	0.93	14			
59	14	4.00 (0.55)	2	0.86	14			
**Translation**								
60	14	4.07 (0.83)	2	0.86	14	4.13 (0.99)	0	1

## Data Availability

The data presented in this study are available on request from the corresponding author.

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
