# Peer review of "Assessing Face Validity of the HexCom Model for Capturing Complexity in Clinical Practice: A Delphi Study"

_healthcare, 2021, doi:10.3390/healthcare9020165_

Round 1

Reviewer 1 Report

Dear authors, to improve your beautiful work, I would like to make a few suggestions for your consideration, most of them regarding to formal presentation:

First I would attach the original version (languages) as supplemental files. I would rewrite bibliographic references; Tables 1 and 3 are offset. I suppose it is an editing problem? Appendix 1 or A? The Likert scale also displaced.

On the other hand, regarding the inclusion criterion of "years of experience" for the group of experts, is it 7 or 8 years minimum?

Finally, isn't gender bias in both groups a limitation? Can work be affected by female predominance?

I would appreciate a more extensive discussion that includes the cost-benefit aspects of implementing the model.

Author Response

Dear reviewer,

We really appreciate your valuable insights and your kind comments. We hope we have managed to response adequately to the following observations:

Point 1: First I would attach the original version (languages) as supplemental files.

Response 1: The original versions of the HexCom instruments and figure have been attached in a compressed as Supplementary Materials 3. These are cited in the manuscript as S3 in the last line of page 2. All supplementary materials have been named in the back matter of the manuscript (page 11).

Point 2: I would rewrite bibliographic references;

Response 2: All bibliographic references have been reviewed and updated in the text. In the reference list, the following have been modified manually: 1-8, 10, 13, 16, 17, 19, 20, 27, 30, 32, 34-37, 43, 47, 49, 51, 55, 59.

Point 3: Tables 1 and 3 are offset. I suppose it is an editing problem?

Response 3: Tables 1 and 3 have been formatted so they do not look offset. I would appreciate it if the editors could make sure these look as neat as possible in the final version.

Point 3: Appendix 1 or A?

Response 3: All in-text references to appendices have been corrected and are now labelled with “A”, “B”, etc. See pages 5 (A) and 6 (B)

Point 4: The Likert scale also displaced.

Response 4: Appendix A (which includes the Likert scale) has been formatted so it does not look offset. I would appreciate it if the editors could make sure this looks as neat as possible in the final version.

Point 5: On the other hand, regarding the inclusion criterion of "years of experience" for the group of experts, is it 7 or 8 years minimum?

Response 5: The right criterion is 7 years, as it is displayed in table 2. In-text description of such inclusion criterion at page 6 has been amended.

Point 6: Finally, isn't gender bias in both groups a limitation? Can work be affected by female predominance?

Response 6: This “bias” reflects how gender is currently distributed in palliative care in Catalonia given that the main inclusion criterion was to have a long experience and be active in palliative care. This clarification has been included in the manuscript, towards the end of page 4 and 3rd paragraph of page 6.

Point 7: I would appreciate a more extensive discussion that includes the cost-benefit aspects of implementing the model.

Response 7: A sentence has been added to the discussion (penultimate paragraph in page 10) and a closing line at the conclusion. Both acknowledge the limitations of implementing the model, particularly in settings beyond home care. These are presented as challenges or lines for future research (e.g. asking non-experts to assess the model) and managerial debates (e.g. discuss adequate resources allocation to allow sufficient time and training to implement the model in highly demanding clinical environments).

Reviewer 2 Report

Dear authors,

your paper presents an approach to capturing complexity in palliative care using HexCom model by using Delphi method. This work is interesting, well-written, and well structured. I think it will be very useful for clinical practice in palliative care.

I will now make some recommendations for improving the quality of the manuscript.

1) Abstract: "a group of 15 experts". However, only 14 professionals accepted to take part in the study

2) Introduction: Is important to explore the particularities of homecare in Spain and the differences between home care and other levels of care. Please clarify this situation.

Who are the professionals in palliative care teams in Spain? Are they just physicians, nurses, social workers, and psychologists? (see 3. and abstract - "representing the full spectrum of healthcare professionals involved in palliative care"

3) Material and methods: 

3.1 - Why have not been included other professionals - physiotherapists, speech therapists, occupational therapists?

3.2 I think table 2 (coordinator Group) is not important for the study field.

3.3 In 2.3.1 - "inclusion criteria of having over 8 years". However in table 3. Characteristics of Expert Panel, the years of palliative experience is from 7 years.

3.4 Table 3. Review the bold: "Age"

4. Discussion: While the strengths associated with the use of HexCom model are clear, I think it would also highlight the obstacles to its uses. To this end, it would be important to highlight the open comments. These open comments are very interesting and useful to think about palliative care, home care, and patient's needs.

Change [43] [64]: [43,64]

change[37] [38] [39]: [37-39]

Change in text: Supplementary 1 &2 for A & B

Change Appendiz 2 - B

Author Response

Dear reviewer,

We really appreciate your valuable insights and your kind comments. We hope we have managed to response adequately to the following observations:

Point 1: Abstract: "a group of 15 experts". However, only 14 professionals accepted to take part in the study.

Response 1: Even though 15 professionals were approached, it is true that only 14 of them accepted to be involved. This has been corrected in the abstract.

Point 2: Introduction: Is important to explore the particularities of homecare in Spain and the differences between home care and other levels of care. Please clarify this situation.

Response 2: In the third paragraph of the introduction, after introducing the systemic approach to palliative care, the case has been made for its particular relevance for palliative home care in Spain given also its coordinating role, in contrast with how palliative care is delivered at other levels of care. A reference (19) has been added to support this claim.

Point 3: Who are the professionals in palliative care teams in Spain? Are they just physicians, nurses, social workers, and psychologists? (see 3. and abstract - "representing the full spectrum of healthcare professionals involved in palliative care"

Response 3: A brief description of palliative care teams has been added to the paragraph following table 1, since it is true that this table ambiguously referred to “specialists”, without further clarification.

Point 4: Why have not been included other professionals - physiotherapists, speech therapists, occupational therapists?

Response 4: In Spain, and particularly in Cataluña, these professionals are very rarely involved in palliative home care. Given that having sufficiently long experience in this field which was the main inclusion criterion of our study, this set of profiles did not meet the criterion. However, we now acknowledge, in the discussion section of the article, that their inclusion must be considered in future research projects.

Point 5: I think table 2 (coordinator Group) is not important for the study field.

Response 5: In accordance with CREDES guidelines on conducting and reporting Delphi Studies, we were aiming to provide a description of the methods as comprehensive as possible. We reckon it is necessary to provide such degree of transparency in as much as this group designed in the questionnaire.

Point 6: "inclusion criteria of having over 8 years". However in table 3. Characteristics of Expert Panel, the years of palliative experience is from 7 years.

Response 6: The right criterion is 7 years, as it is displayed in table 2. In-text description of such inclusion criterion at page 6 has been amended.

Point 7: Table 3. Review the bold: "Age"

Response 7: Table 3 has been formatted again to correct this and other formatting issues.

Point 8: Discussion: While the strengths associated with the use of HexCom model are clear, I think it would also highlight the obstacles to its uses. To this end, it would be important to highlight the open comments. These open comments are very interesting and useful to think about palliative care, home care, and patient's needs.

Response 8: A sentence has been added to the discussion (penultimate paragraph in page 10) and a closing line at the conclusion. Both acknowledge the limitations of implementing the model, particularly in settings beyond home care. These are presented as challenges or lines for future research (e.g. asking non-experts to assess the model) and managerial debates (e.g. discuss adequate resources allocation to allow sufficient time and training to implement the model in highly demanding clinical environments).

Point 9: Change [43] [64]: [43,64] / change[37] [38] [39]: [37-39]

Response 9: All bibliographic references have been reviewed and updated in the text and in the reference list.

Point 10: Change in text: Supplementary 1 &2 for A & B

Response 10: According to the template and the instruction for authors, supplementary material must be labelled as S1, S2, etc.

Point 11: Change Appendiz 2 - B

Response 11: All in-text references to appendices have been corrected and are now labeled with “A”, “B”, etc. See pages 5 (A) and 6 (B)